# I Know Myself Better: Learning Complementary Semantic Views For Self-explaining Cardiovascular Signal Stratification

## Abstract

Explainable artificial intelligence (XAI) offers enhanced transparency by revealing key features, relationships, and patterns within the input data that drive model decisions. In healthcare and clinical applications, where physiological signals serve as inputs to the models for decision making, such transparency is critical for facilitating analysis of inference causality, ensuring reliability, identifying biases, and uncovering new insights. In this work, we introduce a self-explaining multi-view deep learning architecture, that generates task-relevant and human-interpretable masks, attributing feature importance during model inference for stratifying key information from input signals. We implement the 2-view version of the proposed architecture for three clinically-relevant regression and classification tasks related to cardiovascular health, involving electrocardiogram (ECG) or photoplethysmogram (PPG) signals. Experimental results demonstrate that the complementary masks, self-generated by our proposed architecture, outperform well-established post-hoc methods (LIME and SHAP), both qualitatively and quantitatively in explainability. Furthermore, the 2-view model offers task-level performance comparable to or better than the state-of-the-art methods, displaying its broad applicability across various cardiovascular-related tasks. Overall, the proposed method offers new directions for interpretable machine learning and data-driven analysis of cardiovascular signals, envisioning self-explaining models for clinical applications.

## 1 Introduction

Data-driven end-to-end deep learning methods find applications in problems that are difficult to characterize using manually-defined features or traditional statistical analysis. Examples include image classification (He et al., 2016), disease diagnosis (Ronneberger et al., 2015), speech recognition (Abdel-Hamid et al., 2014), and financial market prediction (Fischer & Krauss, 2018). However, the behavior of these models generally lacks transparency, making it difficult to understand how decisions are made by the model from its input.

To enhance the interpretability of deep learning methods, explainable artificial intelligence (XAI) has recently received increased attention, contributing to reliable decision-making (Wang et al., 2020; Hendricks et al., 2016; Ribeiro et al., 2016), analysis of model biases and failure modes (Ras et al., 2018; Karpathy et al., 2016; Geirhos et al., 2018; Vilone & Longo, 2021), and revealing key features, patterns and relationships within input data (Jumper et al., 2021). In healthcare and clinically-relevant applications, where deep learning models are used to make diagnostic decisions from physiological signals (Alberdi et al., 2016; Imani et al., 2016; Castaneda et al., 2018; Seshadri et al., 2019; Betti et al., 2017; El-Hajj & Kyriacou, 2020; Liu et al., 2023; Mousavi et al., 2020; Suresh & Duerstock, 2020), XAI has the potential to identify key patterns in physiological signals that drive decision making, with implications for assessing clinical reliability (Mehta et al., 2023; Chandrasekhar et al., 2020; González et al., 2023), personal health monitoring (Charlton et al., 2022b), efficient distribution of medical resources (Kang & Exworthy, 2022), and continuous monitoring of risk factors in critical care (Rao et al., 2023). However, at present, generalized solutions for interpretable learning from physiological signals are still limited.

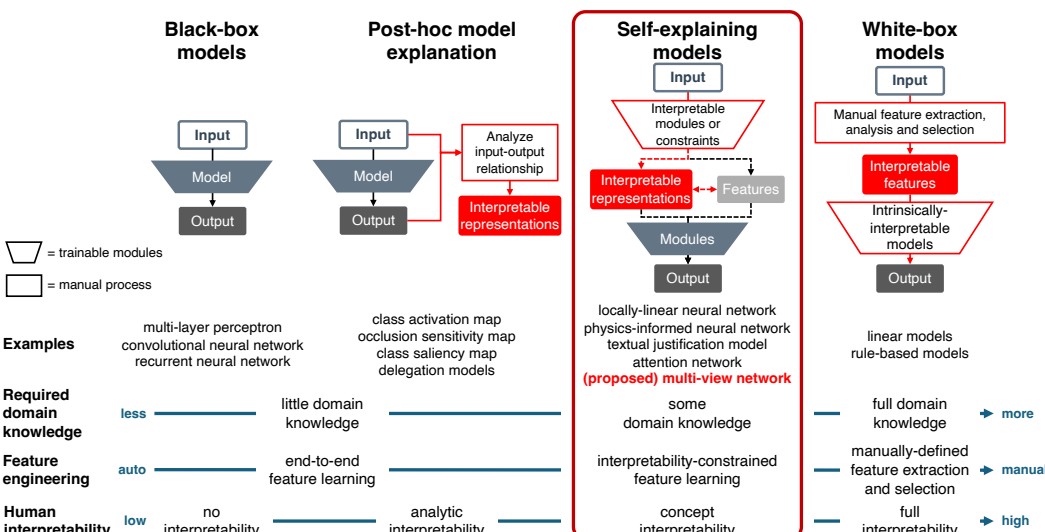

Figure 1: Overview of existing methods for improving interpretability in machine learning models.

In this paper, we introduce a generalized new self-explaining deep learning method that reveals key patterns in cardiovascular signals for stratifying health-related information, with minimum help from prior expert knowledge. Our work offers the following contributions:

- We introduce a generalized approach for learning semantic information from consecutive intervals of an input signal, by attributing each sample in the signal to one of $N$ semantic states. Samples attributed to the same semantic state are expected to form patterns that offer distinct information for making the final clinically-relevant decision.

- We propose a multi-view end-to-end deep learning architecture to implement our learning principles. The proposed architecture includes a mask network that produces multiple mask-modulated versions of the signal, each representing a "semantic view" formed by samples in the signal corresponding to the same semantic states. Each semantic view highlights distinct regions and informative patterns within the signal. Concatenated with an embedding network and a decision network, during supervised training using the task labels, the created semantic views are updated based on the saliency information with respect to the model's output, making the highlighted regions of the signal more relevant to the task. As such, the semantic views can improve model's interpretability through explaining the correspondence between regions in the input signal and the clinically-relevant information inferred from the signal.

- We implement the 2-view version of the proposed multi-view architecture for 3 different classification and regression tasks involving stratifying clinically-relevant information from 2 different cardiovascular signals. We validate the proposed models by quantitatively and qualitatively comparing the correctness of their self-generated explanations, against those created from well-established post-hoc methods, as well as comparing their task-level performance against state-of-the-art methods. We also demonstrate the alignment between the interpretable representations generated by our models, and the domain knowledge from human experts, for each task.

## 2 RELATED WORKS

Figure 1 summarizes existing methods that facilitate interpretability in machine learning models. Unlike white-box models with inherent explainability due to their simplicity, linearity or feature-driven nature, learners in data-driven deep learning models, such as multi-layer perceptron (MLP), convolutional neural network (CNN) or recurrent neural network (RNN), generally lack interpretability when stacked with non-linear activations (Ali et al., 2023). To understand their working principles, post-hoc model explanation methods were developed, to identify the input-output correspondence learned by these models after training. For example, (Zeiler & Fergus, 2014) proposed the occlusion sensitivity analysis for image classification models, to investigate how occluding each

region in the input affects model's output. Alternatively, (Simonyan, 2014) inspected the gradient backpropagated from the class probability to each input pixel, to form a class-specific saliency map that highlights regions in the input that changes the model's output the most. Studies in (Zhou et al., 2016) and (Selvaraju et al., 2017) used the 2-D feature maps generated by the last convolutional layer in CNNs to produce class activation map (CAM) and gradient-weighted CAM (Grad-CAM) that localize regions in the input that are most related to the output. The insights from these approaches were further generalized in later works, such as LIME, DeepLIFT and SHAP (Ribeiro et al., 2016; Lundberg, 2017; Shrikumar et al., 2017), to explain any trained model by finding an interpretable delegation model, with faithfulness to the original non-interpretable model.

Post-hoc model explanation methods explain previously-trained models based on local input-output properties around each sample, which may limit their fidelity in representing the working principles of the original model (Rudin, 2019). Moreover, explanations provided by these methods are not guaranteed to be human understandable, since the models are not regularized to encode interpretable and task-specific *concepts* in the learned representations during training (Alvarez Melis & Jaakkola, 2018; Park & Hwang, 2023). For example, (Troncoso-García et al., 2022) applied the LIME method to a sleep apnea detection model that takes multi-modal inputs (blood pressure (BP), electrocardiogram (ECG), electroencephalogram (EEG), and nasal respiratory signals), however, the LIME method only highlighted a few discrete samples in the input time series, offering limited insight into the key patterns in the signals that characterize sleep apnea.

To address these limitations, recent works focused on developing self-explaining models, with intrinsic interpretability either learned during training or built-in to the model architecture, for offering faithful, stable, and human-understandable explanations. (Alvarez Melis & Jaakkola, 2018) proposed a locally-linear neural network, in which the model is regularized to have local linearity around each sample, for offering inherent explainability. Sharing similar insights, (Sel et al., 2023) proposed a physics-informed neural network (Raissi et al., 2019) for BP estimation, by optimizing an additional physics-based loss to embed physical constrains in the input-output correspondence of the model. Besides, recent advancements in incorporating attention mechanisms in deep learning models (Bahdanau, 2015; Dosovitskiy, 2021; Mousavi et al., 2020; Jin et al., 2021) also enhances model's interpretability, through investigating the attention maps generated by the model that highlights informative patterns or relationships in the model's inputs. Furthermore, the flexibility of learning interpretable representations during model training, enables better human-level understanding of the explanations produced by the model. For example, (Hendricks et al., 2016) considered joint training of classification and language models for image classification tasks, to generate human-understandable explanations to the produced classifications in natural language.

Due to the unique capabilities of self-explaining models, we seek to develop generalized solutions that support healthcare decisions, by improving human understanding of health-related inference from input signals.

## 3 METHODS

### 3.1 PROBLEM FORMATION

Let $\mathbf{S} = \{\mathbf{x}_1, \mathbf{x}_2, \cdots, \mathbf{x}_T\} = \{\mathbf{x}_t\}_{t=1}^{T}$ denotes a continuous multivariate time series interval with $T$ samples, where $\mathbf{x}_t = [x_{1,t}, \cdots, x_{D,t}] \in \mathbb{R}^D$ is a $D$-dimensional sample at time instant $t$. $\mathbf{S}$ is labeled by $y$ denoting the clinically-relevant information to be inferred from the signal. As such, the stratification task can be defined in general as producing $\tilde{y}$, an estimation of $y$, from $\mathbf{S}$.

To enable generalized and interpretable estimations, we build on prior work (Wang et al., 2011; Yue et al., 2022; Deldari et al., 2021; Gharghabi et al., 2019), which assumes that the input signals reflect the behavior of their underlying system, whose dynamics are describable by a number of latent semantic states closely related to $y$. To facilitate human-level interpretation and segment time series into multiple distinct regions, each sample $\mathbf{x}_t$ in the time series $\mathbf{S}$ is attributed to one and only one of the $N$ semantic states $\mathbf{u}_1, \mathbf{u}_2, \cdots, \mathbf{u}_N$. Samples sharing the same $n^{th}$ semantic state form a sub-series $\mathbf{s}_n$, referred to as a semantic view reflecting distinct characteristics of $\mathbf{S}$ relevant to $y$. Therefore, through extracting information from all $N$ semantic views, one can retrieve features from the time series $\mathbf{S}$ that comprehensively describe the characteristics of the underlying system, for

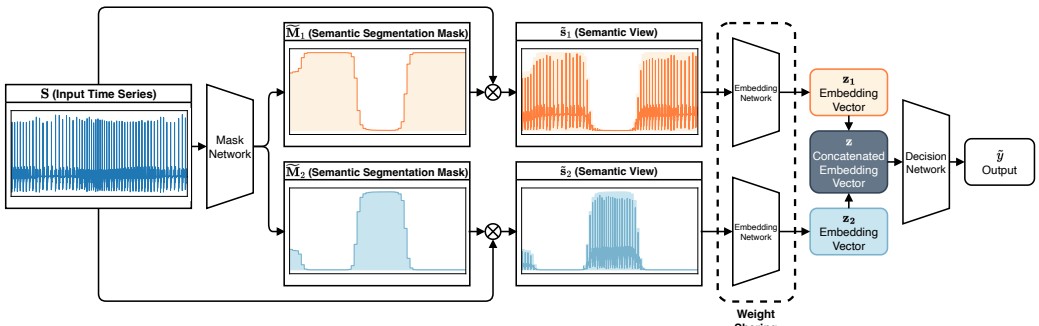

Figure 2: The proposed multi-view model architecture for self-explaining deep learning. An example of creating two views, as used in the experiments in this study, is shown. A mask network is trained to form complementary semantic views from the input signal. A shared embedding network is used to extract features from each semantic view. A decision network combines features extracted from all semantic views to form the final output.

estimating the desired information $y$. We thereby describe the general process of signal stratification in 3 steps:

- **Semantic segmentation:** Attribute each sample $\mathbf{x}_t$ in $\mathbf{S}$ to one of the $N$ semantic states, yielding $N$ semantic views, $\mathbf{s}_1 = \{\mathbf{x}_t | \mathbf{x}_t \in \mathbf{u}_1\}, \cdots, \mathbf{s}_N = \{\mathbf{x}_t | \mathbf{x}_t \in \mathbf{u}_N\}$.

- **Embedding extraction:** Learn a low-dimensional embedding representation $\mathbf{z}_n$, from each of the high-dimensional semantic views $\mathbf{s}_n$. As such, $\mathbf{z}_1, \cdots, \mathbf{z}_N$ are expected to form complete representations of all distinct semantic information that $\mathbf{S}$ carries.

- **Decision:** Form a final output $\tilde{y}$ based on embeddings extracted from all semantic views.

Essentially, the semantic segmentation process is equivalent to performing a $N$-class sample-by-sample classification on $\mathbf{S}$. However, unlike its conventional form, where the segmentation is learned under the supervision of manual annotations (Peimankar & Puthusserypady, 2021; Moskalenko et al., 2020), here, no prior implication is specified to each of the $N$ semantic states. During training, the model is left to spontaneously learn how samples in $\mathbf{S}$ should be attributed to each semantic state, such that the patterns retained in each semantic view are most informative for optimizing the estimation of $y$. As such, the semantic views produced by the model during inference, can explain informative patterns in $\mathbf{S}$ that drives the estimation of $y$.

## 3.2 PROPOSED METHOD

We propose a multi-view model architecture to implement the abovementioned semantic segmentation, embedding extraction, and decision procedures, within a unified end-to-end deep learning framework (Figure 2 displays a 2-view implementation).

### 3.2.1 LEARNING FOR SEGMENTATION

For interpreting and optimizing each semantic view with deep learning models, we form unified semantic views $\widehat{\mathbf{s}}_n$ by padding $\mathbf{s}_n$ with zero vectors to the same length as $\mathbf{S}$, following

$$\widehat{\mathbf{s}}_n = \{\widehat{\mathbf{x}}_t\}_{t=1}^T, \ \widehat{\mathbf{x}}_t = \begin{cases} \mathbf{x}_t & \text{if } \mathbf{x}_t \in \mathbf{u}_n \\ \mathbf{0} & \text{otherwise} \end{cases}. \tag{1}$$

As such, each semantic view $\widehat{\mathbf{s}}_n$ is derived, by applying a segmentation mask $\mathbf{M}_n$ to the original signal $\mathbf{S}$ through calculating element-wise multiplication, as

$$\mathbf{M}_n = \{m_{n,t}\}_{t=1}^T, \ m_{n,t} = \begin{cases} 1 & \text{if } \mathbf{x}_t \in \mathbf{u}_n \\ 0 & \text{otherwise} \end{cases}, \tag{2}$$

$$\widehat{\mathbf{s}}_n = \mathbf{S} \otimes \mathbf{M}_n = \{m_{n,t} \times \mathbf{x}_t\}_{t=1}^T.$$

Since each sample in the time series $\mathbf{S}$ is attributed to one and only one semantic state, the masks $\mathbf{M}_1, \cdots, \mathbf{M}_N$ are complementary to each other. To enable automatic learning of these segmentation masks through deep learning and gradient-based optimization, we use a softmax activation function to facilitate the complementary constraints among $N$ semantic views, following

$$\widetilde{\mathbf{M}}_n = \{p_{n,t}\}_{t=1}^T, \ p_{n,t} = \frac{e^{h_{n,t}}}{\sum_{n=1}^N e^{h_{n,t}}}, \tag{3}$$

where $h_{n,t}$ are the logit outputs of the mask network for each semantic state $\mathbf{u}_n$ and sample $\mathbf{x}_t$, and $p_{n,t}$ are the normalized probabilities that attribute each sample at $t = 1, \cdots, T$, to each semantic state $n = 1, \cdots, N$. As such, $\widetilde{\mathbf{M}}_n$ can be learned from $\mathbf{S}$ using the mask network. These learned masks are applied to $\mathbf{S}$ itself to create $N$ semantic views $\widetilde{\mathbf{s}}_n$ to be optimized by the subsequent embedding and decision networks, as

$$\widetilde{\mathbf{s}}_n = \mathbf{S} \otimes \widetilde{\mathbf{M}}_n = \{p_{n,t} \times \mathbf{x}_t\}_{t=1}^T. \tag{4}$$

Applying $\widetilde{\mathbf{M}}_n$ to $\mathbf{S}$ thereby retains samples in $\mathbf{S}$ attributed to the semantic state $\mathbf{u}_n$ with high amplitudes, and attenuates the remaining samples that correspond to other semantic states, in $\widetilde{\mathbf{s}}_n$. To enable learning of informative patterns from consecutive segments of the input signal, one can enforce a minimum duration $L$ for each semantic mask, ensuring consecutive samples within the mask share the same semantic state and $\{p_{n,t}\}$ values. We utilize this approach for ECG and PPG signals in our experiments, as explained in Appendix A.2.

During backpropagation, each semantic view is updated, through optimizing the segmentation masks $\widetilde{\mathbf{M}}_n$ for fitting the model's output $\tilde{y}$ to $y$. The gradient of each element in $\widetilde{\mathbf{M}}_n$ with respect to the loss function $f$ evaluated between $\tilde{y}$ and $y$ is

$$\frac{\partial f(\tilde{y}, y)}{\partial p_{n,t}} = \frac{\partial f(\tilde{y}, y)}{\partial \tilde{y}} \times \sum_{d=1}^D \frac{\partial \tilde{y}}{\partial \tilde{s}_{n,d,t}} \times x_{d,t}, \tag{5}$$

where $\frac{\partial \tilde{y}}{\partial \tilde{s}_{n,d,t}}$ is the saliency map of the embedding and decision networks in the model that localize samples in each semantic view with the greatest effect on the model's output (Simonyan, 2014; Selvaraju et al., 2017). Such saliency information in the gradient can thereby facilitate inclusion of different informative patterns in the input signal in each semantic view during optimization, which drive the final decision of the model.

Overall, the mask network in the proposed muti-view deep learning architecture offers self-explainability, through creating and optimizing multiple semantic views $\widetilde{\mathbf{s}}_1, \cdots, \widetilde{\mathbf{s}}_N$ from the input. Each semantic view forms a different interpretable perspective of $\mathbf{S}$, highlighting characteristic patterns in the signal that provides semantic information toward the output. Meanwhile, all complementary semantic views together retain all samples in $\mathbf{S}$, to ensure comprehensive feature extraction from $\mathbf{S}$ for making the final decision.

### 3.2.2 Learning For Making Decisions

An embedding network is employed to encode a low-dimensional embedding representation $\mathbf{z}_n$, for each semantic view $\widetilde{\mathbf{s}}_n$ created by the mask network. Inspired by the use of shared encoders for learning representations from augmented views (Zagoruyko & Komodakis, 2015; Chen et al., 2020; Yue et al., 2022; Yang et al., 2022b; Deldari et al., 2021), we employ weight sharing in the embedding network across all semantic views. This approach enables the learning of generalized filters and ensures informative gradients are propagated to all semantic states, such that the mask network can learn to segment the input signal properly. Moreover, weight sharing also ensures feature comparability across semantic views, enabling the decision network to distinguish and extract comparative features from different semantic states (Wang et al., 2024; Schlesinger et al., 2020).

Based on the embedding vectors $\mathbf{z}_1, \cdots, \mathbf{z}_N$ extracted comprehensively from all semantic views, the decision network in the model is trained to form the final output $\tilde{y}$ that estimates the task label $y$. For general tasks, a simple concatenation of all embeddings $\mathbf{z} = [\mathbf{z}_1, \cdots, \mathbf{z}_N]$ forms the input of the decision network.

Overall, the proposed multi-view deep learning framework combines the mask, embedding, and decision networks together, as one unified deep learning model trained under single supervision of the task label $y$, for self-explaining physiological signal stratification.

## 4 EXPERIMENTS

For validation of usability and interpretability of the proposed architecture, we considered 3 different tasks for stratifying clinically-relevant information (obstructive sleep apnea (OSA) detection (classification), heart rate variability (HRV) estimation (regression), and BP elevation ($\Delta$BP) detection (classification), from 2 cardiovascular signals: the ECG, and the photoplethysmogram (PPG). These two cardiovascular signals are characteristically different in waveform morphology and the physiological information they provide. Although our model architecture allows choosing an arbitrary number of semantic states ($N$) for different granularity of segmentation and interpretability, here we focus on $N = 2$ to highlight the most discernible patterns in the signal that deliver different information for optimal human understandability, and leave explorations of other settings for future studies. Detailed descriptions of each task as well as information of datasets used for validation can be found in Appendix A.1. The hyperparameter settings used for implementing the 2-view model for each task are summarized in Appendix A.2.

## 5 RESULTS AND DISCUSSIONS

### 5.1 QUANTITATIVE INTERPRETABILITY ANALYSIS

For each task, we first quantitatively compare the correctness of self-generated explanations from our proposed 2-view model against those created from two well-established post-hoc methods, LIME and SHAP.

- **LIME** (Ribeiro et al., 2016) considers a linear surrogate model that maps the presence or absence of interpretable elements, encoded as binary vectors, to the local outputs of the explained model around a particular input signal, such that the linear coefficients of the fitted model attribute the importance of each element in the input signal toward model's output.

- **SHAP** (Lundberg, 2017) uses the Shapley value (Shapley, 1953) to evaluate the importance of each element in the input signal, which assesses changes in model's output when trained on different subsets of input elements, including or withholding the attributed element. SHAP approximates Shapley values using various methods (Ribeiro et al., 2016; Shrikumar et al., 2017; Štrumbelj & Kononenko, 2014). Here, we used the Gradient SHAP method (Lundberg, 2017).

To evaluate the quality of semantic segmentation masks generated by the 2-view model as interpretable representations, we treat the amplitudes of each of the two masks (denoted as mask 1 and mask 2) as feature attribution weights that rank the importance of each length-$L$ window in the input signal toward model's output. The interpretations self-generated by the 2-view model, are then compared with feature attributions created on the same model, through LIME and SHAP.

For an objective quantification of interpretability, we used the well-established incremental deletion method (Petsiuk, 2018; Nauta et al., 2023; Samek et al., 2016). This method evaluates how incrementally perturbing important input features, identified by high mask amplitudes of the 2-view model or large absolute values of attribution scores by LIME or SHAP, impacts the model's output. We investigated how perturbing input signal windows affects the 2-view model's test performance, to evaluate the correctness and sensitivity of window importance suggested by different explanation methods. Additionally, a baseline is considered by perturbing each window in the input signal at a random sequence.

Figure 3 summarizes the incremental deletion curves evaluated on the three 2-view models trained for each of the considered tasks. A lower area under deletion curve (AUDC) indicates better explanations that highlight essential regions in the signal closely related to model's decision.

Across all tasks, LIME and SHAP outperformed the baseline, with SHAP having improved performance over LIME due to its better adherence to desirable properties of model explainers (Lundberg, 2017). Meanwhile, for the proposed 2-view model, one (HRV task) or both (OSA and $\Delta$BP tasks) self-generated masks offered optimal AUDC performance over the post-hoc and baseline methods. This superior performance can be attributed to the multi-view network's architecture, which uses the created masks to modulate the inputs to the embedding and decision networks, ensuring a straightforward correspondence between the interpretations and the model's output. Interestingly, for the

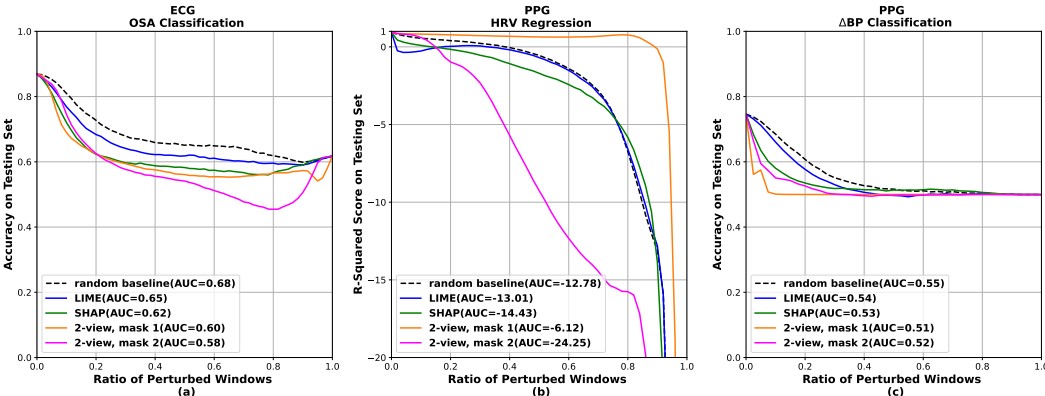

Figure 3: Evaluation of testing performance of the 2-view model in incremental deletion tests, using window importance suggested by LIME, SHAP, and each of the 2 semantic segmentation masks self-created by the proposed model for each of the considered tasks. A lower area under the deletion curve (AUDC) implies that the corresponding explanation method provides more accurate attributions of the signal regions that drive the model's decision.

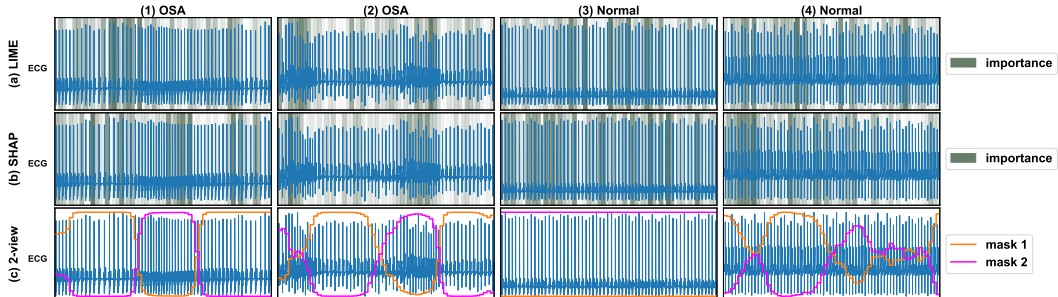

Figure 4: Rows (a) and (b): absolute attribution scores generated by LIME and SHAP, respectively, for explaining the 2-view model trained for ECG-based OSA classification task. Row (c): self-generated semantic segmentation masks from the 2-view model itself. Columns (1) and (2) represent examples of OSA condition, and columns (3) and (4) represent examples of normal condition. Deeper color indicates higher importance.

HRV regression task, it can be observed that mask 1 performs worse than the random baseline, which can be due to two factors. First, Equation (3) constraints samples with high amplitudes in one mask to correspond to low amplitudes in the other, causing mask 1 to highlight regions of reversed importance relative to mask 2. Second, tasks relying primarily on one semantic view limit the influence of the other mask. Figure 3 shows that OSA and $\Delta$BP tasks use both views, while the HRV task depends mainly on the view modulated by mask 2. As will be seen, these align with qualitative analysis (Section 5.2) and clinical knowledge related to each task.

Additionally, we should state that the self-generated feature attribution is more computationally efficient than LIME and SHAP, since the masks are retrieved through single model inference on the evaluated input signal. Comparatively, both LIME and SHAP run model inferences repeatedly on numerous augmented samples around the input signal to capture the model's behavior, thereby requiring higher computational budget.

## 5.2 QUALITATIVE INTERPRETABILITY ANALYSIS

We now present and discuss the semantic views generated by the proposed 2-view model for each task qualitatively, in comparisons with attributions created by SHAP and LIME.

### 5.2.1 OBSTRUCTIVE SLEEP APNEA (OSA)-TASK: CLASSIFICATION, INPUT: ECG

Figure 4 summarizes examples of interpretations of the 2-view model trained for OSA classification, generated through LIME, SHAP, and the semantic masks of the 2-view model itself.

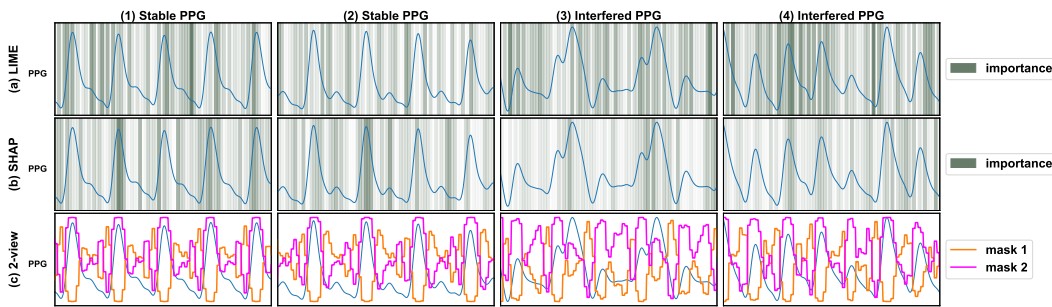

Figure 5: Rows (a) and (b): absolute attribution scores generated by LIME and SHAP, respectively, for explaining the 2-view model trained for PPG-based HRV regression task. Row (c): self-generated semantic segmentation masks from the 2-view model itself. Columns (1) and (2) represent examples of stable PPG, and columns (3) and (4) represent examples of interfered PPG. Deeper color indicates higher importance.

Clinical studies have found OSA to be characterized by cyclical variation of the heart rate (CVHR) in the ECG signal (Guilleminault et al., 1984; Hayano et al., 2011). From Figure 4, one can see that the segmentation masks generated by the proposed 2-view model clearly capture such information. Specifically, for examples labeled as OSA (columns (1)-(2) in Figure 4), segments corresponding to high heart rate (HR) (manifested as dense peaks in ECG) are attributed to mask 2. Consequently, segments with low HR (manifested as sparse peaks in ECG) are retained by mask 1. Over time, the dominant semantic state showing the highest probability swaps between the two states corresponding to low-HR and high-HR for multiple times in OSA examples, matching the characteristics of CVHR corresponding to OSA. This explains the AUDC performance in Figure 3(a) when regions are deleted based on the importance scores from either mask 1 or mask 2, as both are essential for capturing CVHR. Although LIME and SHAP also capture some CVHR properties (subfigures (a1), (b1), (a2) and (b2) of Figure 4), they do not localize consecutive regions with high or low HR, or the occurrence of HR changes, as good as the semantic masks created by the 2-view model, resulting in their inferior AUDC performance in Figure 3(a). Meanwhile, in normal examples (columns (3)-(4) in Figure 4), due to the lack of CVHR pattern in the signal, the model either consistently suggests highest probability for a single semantic state over time (subfigure (c3) of Figure 4), or shows uncertainties in distinguishing between semantic states (subfigure (c4) of Figure 4).

### 5.2.2 HEART RATE VARIABILITY (HRV)-TASK:REGRESSION, INPUT: PPG

Figure 5 summarizes examples of window importance in stable and interfered PPG signals, evaluated by LIME, SHAP, and the semantic masks of the 2-view model that is trained for the HRV regression task.

PPG-based HRV metrics are derived by calculating the variability of inter-beat interval (IBI), which requires the deep learning model to locate occurrence of cardiac cycles in the PPG signal. In Figure 5, mask 2 of the 2-view model clearly and steadily captures this feature among cardiac cycles, for PPG signals with both stable (columns (1)-(2) in Figure 5) and interfered (columns (3)-(4) in Figure 5) morphologies, respectively. This explains the superior AUDC performance of mask 2 in Figure 3(b), since it highlights the most essential feature (peaks of PPG indicating cardiac cycles) for accurate HRV evaluation. Meanwhile, mask 1 highlights other regions in the PPG signal with weak relevance to HRV, resulting in the worst AUDC performance. Comparatively, LIME and SHAP have very limited ability to locate PPG cycles related to HRV estimation. For SHAP, although it highlights some PPG beats (subfigures (b1)-(b4) of Figure 5), it lacks the beat-to-beat stability seen in the self-created semantic mask from the 2-view model, which captures all beats in the signal.

### 5.2.3 BLOOD PRESSURE ELEVATION (ΔBP)-TASK: CLASSIFICATION, INPUT: PPG

Figure 6 illustrates two examples of PPG and its derivatives for each of the normal and elevated BP conditions, along with channel-specific model interpretations generated by LIME, SHAP, and the semantic masks from the 2-view model, trained on the BP-elevation detection classification task.

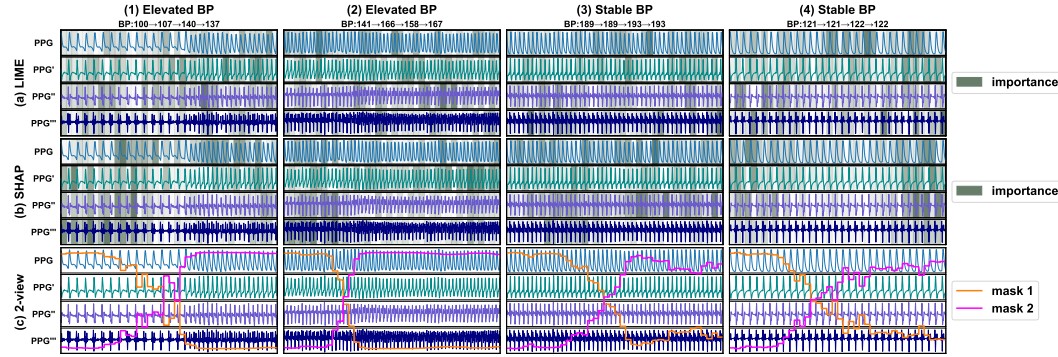

Figure 6: Rows (a) and (b): channel-specific absolute attribution scores generated by LIME and SHAP, respectively, for explaining the 2-view model trained for PPG-based ΔBP classification task. Row (c): self-generated semantic segmentation masks from the 2-view model itself. Columns (1) and (2) represent examples corresponding to elevated BP, and columns (3) and (4) represent examples corresponding to stable BP. Deeper color indicates higher importance.

Within a given duration, BP elevation occurs when higher BP values follow lower ones. Following this definition, the 2-view model divides the earlier and later regions of the PPG signal into different semantic states that potentially correspond to baseline and elevated BP (Figure 6). This property is also partially captured by SHAP in subfigures (b2) and (b4), but not as clear as the self-created semantic masks from the 2-view model, where the transition from one to the other semantic state clearly locates a potential change point. In subplot (c1) of Figure 6, the masks precisely locate the instance where major changes in PPG signal's morphology and its peak-to-peak interval take place. In subplots (c2), although no apparent changes are seen in PPG or its first derivative (PPG'), the mask locates the instance of minor changes in the patterns of second and third derivatives (PPG'' and PPG'''), supporting the observation that certain BP-related information is only present in the higher-order derivatives of the PPG signal (Gupta et al., 2022). Since it would be necessary to extract information before and after BP elevation to characterize the level of elevation, regions retrained by high amplitudes in both masks would be essential for driving model's output, which explains the low AUDC values observed when using either mask 1 or 2 in Figure 3(c).

## 5.3 TASK-LEVEL PERFORMANCES ANALYSIS

Table 1 compares the task-level regression and classification performance of the proposed 2-view model for each of the 3 considered tasks, with results from task-specific state-of-the-art methods. Additionally, from the 2-view model, a basic end-to-end deep learning model is configured for each task by removing the mask network, and extracting a single embedding vector directly from the input signals using the same embedding network, to infer the outputs. All 2-view and ablation models are trained for estimating the labels of each task from scratch.

For the OSA and ΔBP classification tasks, the proposed models were compared to prior deep learning models, using the same dataset for training and testing. For the HRV regression task, results from the proposed model were compared against direct pulse rate variability (PRV) estimates from the PPG signal, obtained using widely-accepted beat-detection algorithms (QPPG (Vest et al., 2018) and ERMA (Elgendi et al., 2013)), benchmarked in (Charlton et al., 2022a) and used in state-of-the-art PPG-based HRV studies (Mejía-Mejía et al., 2022; Guichard et al., 2024).

From Table 1, it can be seen that the proposed 2-view models show comparable or better results compared to the state-of-the-art methods and the ablation models, while also offering self-explainability. The 2-view model outperforms the ablation model through training a shared embedding network to learn from two mask-modulated versions of the input signals. This suggests the effectiveness of leveraging complementary perspectives for learning from time series data.

It is worth noting that the state-of-the-art models (Yang et al., 2022a; Yeh et al., 2022; Shen et al., 2021; Chang et al., 2020) are distinctively different in architectures, while some solutions (Yang et al., 2022a; Shen et al., 2021) require manual extraction of task-specific features from the input signal before deep learning models can be applied. Consequently, a model solution for one task

Table 1: Performance comparison of the proposed approach against task-specific state-of-the-art methods and ablation models, for each cardiovascular-relevant task with ECG or PPG as inputs.

| ECG OSA Classification | | | PPG HRV Regression | | | | PPG ΔBP Classification | | |
|---|---|---|---|---|---|---|---|---|---|
| Methods | ACC↑ | AUC↑ | Methods | MNN | MAE↓ SDNN | RMSSD | Methods | ACC↑ | F1↑ |
| SEResGNet (Yang et al., 2022a) | 0.903 | 0.965 | PRV from QPPG (Guichard et al., 2024) | 6.226 | 7.598 | 11.398 | ΔBP-Net (Wang et al., 2024) | 0.760 | 0.751 |
| CNN+Wavelet (Yeh et al., 2022) | 0.886 | - | PRV from ERMA (Mejía-Mejía et al., 2022) | 4.706 | 5.457 | 8.099 | | | |
| MSDA-CNN (Shen et al., 2021) | 0.894 | 0.964 | | | | | | | |
| CNN (Chang et al., 2020) | 0.879 | 0.94 | | | | | | | |
| (Proposed) 2-view | 0.869 | 0.939 | (Proposed) 2-view | 3.295 | 2.406 | 2.966 | (Proposed) 2-view | 0.746 | 0.729 |
| (Ablation) Remove Mask Network | 0.827 | 0.904 | (Ablation) Remove Mask Network | 3.054 | 2.538 | 3.091 | (Ablation) Remove Mask Network | 0.672 | 0.665 |

may not be applicable to other tasks involving different types of signals. In contrast, the proposed 2-view model has shown to work with two distinct cardiovascular signals with differing waveform morphologies and physiological information, on three diverse tasks, including both classification and regression, with minimal to no performance compromise, demonstrating its broad applicability.

It should be noted that the proposed model has the potential to enhance task-level performance. As a proof-of-concept study, here, we used very basic deep learning architectures (CNN, MLP, and long-short term memory (LSTM)) to highlight the architectural design of our model for enabling self-explainability. Replacing modules in the current model with more advanced alternatives, could further enhance classification and regression performance. As an example, we found that substituting the 1-D modified ResNet blocks (He et al., 2016) in the embedding network of the 2-view model (shown in Figure 7(b)) with 1-D modified Res2Net blocks (Gao et al., 2019), can further improve the testing performance of PPG-based ΔBP classification, to ACC= 0.751 and F1= 0.739.

## 6 LIMITATIONS

While we proposed a multi-view architecture for self-explainability, as a proof-of-concept, our experiments were limited to a 2-view configuration and focused on cardiovascular signals. The proposed model, however, has the potential to be extended to configurations with more views to uncover hidden insights, and to be applied to a broader range of signal types or domains, which are left for future studies. Furthermore, there are potentials to improve explainability. In the current multi-view model, the embedding and decision networks lack integrated interpretability, thus making it hard to quantify the correspondence between each semantic view and the model's output directly. Using alternative architectures for the embedding network, or considering domain-agnostic model interpretation methods in combination with the semantic masks, may further improve the interpretability of the proposed model.

## 7 CONCLUSIONS

Self-explaining models provide unique opportunities for understanding the working principles of deep learning models. To facilitate self-explaining learning from input signals, we introduced a generalized form for learning distinct semantic information from continuous intervals, and proposed a generalized multi-view deep learning architecture that creates multiple complementary semantic views from the input signal for enhanced interpretability and feature extraction. Trained under the supervision of the task label only, the model optimizes its semantic views through the saliency of embedding and decision networks, achieving interpretability by highlighting input patterns that convey relevant physiological information. Tested on 3 real-world cardiovascular signal stratification tasks with 2 different signals, the feature attributions self-created by 2-view implementation of the proposed model outperforms post-hoc model explanation methods both quantitatively and qualitatively, providing clearer explanations of patterns in the input signal that drive decisions, while achieving task-level classification and regression performance comparable to or better than task-specific state-of-the-art methods. Overall, we expect the proposed multi-view framework to enhance data-driven, interpretable analysis of physiological signals, advancing self-explaining models for accurate, reliable computer-aided diagnosis and health monitoring in clinical applications.

## 8 REPRODUCIBILITY STATEMENT

The code base, datasets, and trained models used for producing results summarized in Table 1 and Figures 4, 5, and 6 are available from Kaggle at `https://www.kaggle.com/datasets/anonymous6bg09hn/n4txg4xmtuyj`.

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

## A APPENDIX

### A.1 TASKS AND DATASETS INFORMATION

**1) ECG-based OSA detection:** OSA is a sleep breathing disorder with major negative effects on sleep quality, leading to fatigue, hypertension, cerebrovascular complications, and sudden deaths (Javaheri et al., 2017). Traditional OSA diagnosis requires overnight monitoring and manual scoring of multiple signals at sleep labs, which is obtrusive, time consuming, and costly. Recent studies on ECG-based OSA detection using machine learning envisions timely and automatic OSA diagnosis outside hospital, through compact wearable devices (Liu et al., 2023; Yang et al., 2022a; Chang et al., 2020). The Apnea-ECG database (Penzel et al., 2000) was considered in this study for OSA detection, to form a binary classification task of attributing each 60-s ECG segment in the dataset as "apnea" or "normal". The proposed models are trained and tested on the predefined partition of $17,023$ "released" and $17,248$ "withheld" segments in the dataset, for a fair comparison with recently-proposed SOTA solutions validated using the same dataset and train-test partition (Yang et al., 2022a; Yeh et al., 2022; Shen et al., 2021; Chang et al., 2020).

**2) PPG-based HRV estimation:** Evaluation of ultra-short term HRV within 10-s intervals has found emerging applications in assessing mental stress (Can et al., 2019; Landreani et al., 2017), cardiovascular risk factors (Kang et al., 2022), and cognitive functions (Mahinrad et al., 2016). However, acquiring gold-standard HRV from the ECG signal can have limitations in certain scenarios, and the PPG signal has been considered as an ideal and low-cost surrogate for HRV estimation (Georgiou et al., 2018; Mejía-Mejía et al., 2021). In this study, we formed training, validation and testing sets consisting of $524,868$, $61,676$, and $74,736$ 10-s ECG and PPG segments from the "Training" and "Calibration-free testing" partitions of PulseDB (Wang et al., 2023). We extracted ground-truth HRV metrics (mean normal-to-normal interval (MNN), standard deviation of normal-to-normal interval (SDNN), and root mean square of successive interval differences (RMSSD)) from each 10-s segment of ECG, and evaluated the regression performance of estimating the same metrics from only the PPG signal recorded simultaneously with the ECG.

**3) PPG-based BP elevation detection:** Hypertension is a leading cause of death. PPG-based tracking of changes in BP ($\Delta$BP) is essential for non-invasive and unobtrusive identification of hypertensive emergencies (Wang et al., 2024), which also forms the basis of cuff-less BP estimation (Stergiou et al., 2023) for continuous tracking of cardiovascular risk factors. In this study, we used the same training, validation, and testing partitions of PulseDB (Wang et al., 2023) used in (Wang et al., 2024), with $202,954$, $23,718$ and $23,684$ training, validation and testing samples, to evaluate the accuracy of detecting abrupt systolic BP (SBP) elevations from the PPG signal. In consistence with (Wang et al., 2024), we considered a binary classification task of identifying the presence of acute SBP elevation greater than 10mmHg, within 40-s intervals of PPG as well its first to third order derivatives.

For the OSA detection task, following (Chang et al., 2020), the ECG signal was band-pass filtered using a 4th-order Butterworth filter between 0.5 and 15 Hz. For all tasks, all ECG and PPG signals were resampled to 125 Hz, and linearly remapped between 0 and 1 within the segment used as the input of the model, for unified machine learning using the proposed model.

## A.2 MODEL IMPLEMENTATION AND HYPERPARAMETER SELECTION

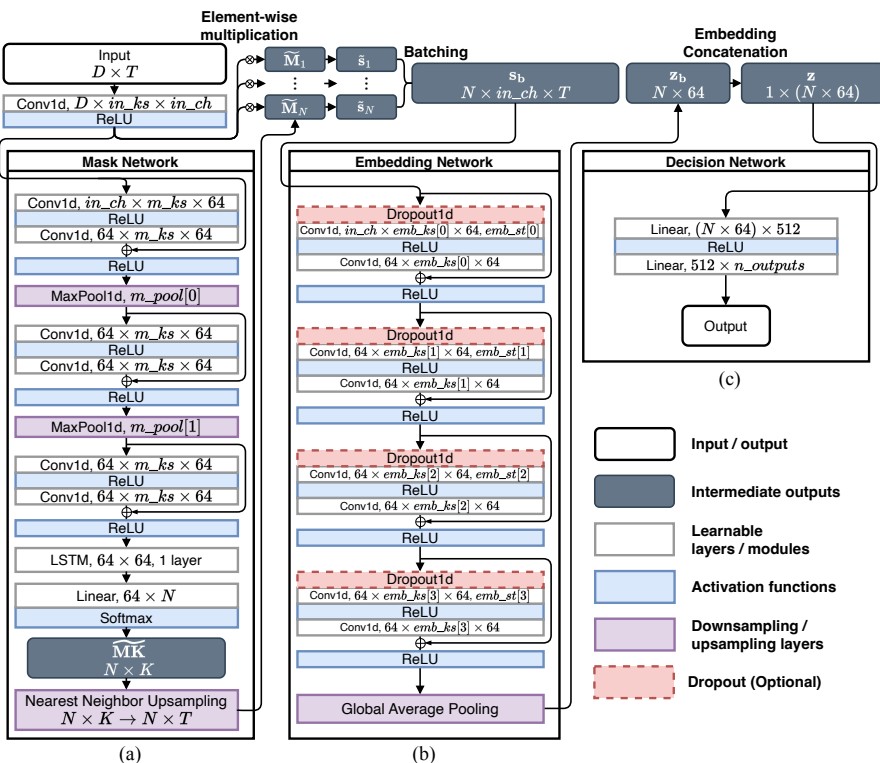

Figure 7: Layer-wise implementation of the proposed multi-view self-explaining deep learning architecture for stratifying ECG and PPG signals in 3 different tasks. Different tasks were fulfilled with different configurations of hyperparameters. (a): The mask network. (b): The embedding network. (c): The decision network.

Figure 7 depicts the layer-wise implementation of the multi-view model used for addressing all 3 tasks, following the architecture introduced in Figure 2. For the mask network, a CNN-LSTM architecture is considered for modeling the transitions between different semantic states over time. For extracting low-dimensional embedding from each semantic view, a hierarchical CNN with ResNet backbone (He et al., 2016) is considered for simplicity, with a final global average pooling (GAP) layer in the network for reducing the temporal dimension of the embedding to 1, which has shown to help enforces correspondence between each semantic view and the extracted embedding (Lin et al., 2014), and facilitate inclusion of all related regions in the signal in each semantic view (Zhou et al., 2016). Finally, a fully-connected decision network taking the concatenation of embedding vectors from all semantic views as input, is used to generate the final output of the model.

To facilitate learning of informative patterns from the ECG and PPG waveform, we enforce a minimum duration $L$, within which consecutive samples should be attributed to the same semantic state,

$$
\begin{aligned}
&\forall \mathbf{x}_t \in \mathbf{u}_n, \ \exists i, L, \ \text{s.t. } L > 0, \ 0 \le i \le L, \\
&\{\mathbf{x}_{t-i}, \mathbf{x}_{t-i+1}, \cdots, \mathbf{x}_{t-i+L}\} \subseteq \mathbf{u}_n,
\end{aligned}
\tag{6}
$$

such that the embedding and decision networks in the model learn from patterns formed by at least $L$ consecutive samples in $\mathbf{S}$ in any semantic view, which prevents the model from creating semantic views that overfit to individual samples in the signal, not corresponding to informative patterns.

In practice, (6) is implemented by evenly placing piecewise-constant windows $\mathbf{W}_{n,1}, \cdots, \mathbf{W}_{n,K}$ in the learned segmentation masks $\widetilde{\mathbf{M}}_n$. All $p_{n,t}$ within each window $\mathbf{W}_{n,k}$ are assigned to the same value $p_{n,k}$, such that

$$\mathbf{W}_{n,k} = \{p_{n,t}\}_{t=(k-1)\times L+1}^{k\times L}, \; p_{n,t} = p_{n,k}, \; \forall p_{n,t} \in \mathbf{W}_{n,k}. \tag{7}$$

This is facilitated by using maxpooling and nearest neighbor upsampling in the mask network, shown in Figure 7(a). Maxpooling not only enlarges the reception field for learning global information from the input for generating segmentations, but also produces the mask kernels $\widetilde{\mathbf{MK}} \in \mathbb{R}^{N \times K}$, that learns $p_{n,k}$ for each window $\mathbf{W}_{n,k}$. Then, the nearest neighbor method was used to upsample $\widetilde{\mathbf{MK}}$ to $\widetilde{\mathbf{M}}_n$, fulfilling Equation (7). Consequently, $T = K \times L$, and $L$ equals to the total downsampling factor of the maxpooling layers in the mask network, before the upsampling takes place.

For each task, the same model architecture in Figure 7 was implemented, while different combinations of hyperparameters were selected manually for optimized performances and interpretability. Table 2 summarizes the hyperparameter configurations used for each task.

Table 2: Summary of hyperparameter and model training settings used for each of the 3 cardiovascular-related signal stratification tasks. BCE: binary cross entropy. MSE: mean squared error.

| Parameter | Explanation | ECG OSA Classification | PPG HRV Regression | PPG $\triangle$BP Classification |
|---|---|---|---|---|
| $N$ | Number of semantic views created by network | 2 | 2 | 2 |
| $D$ | Dimension of each sample in input signal | 1 | 1 | 4 |
| $T$ | Number of samples in the input time series interval | 7500 | 1250 | 5000 |
| $L$ | Duration of each piece-wise constant window in each learned mask | 125 | 5 | 125 |
| $K$ | Number of piece-wise constant windows in each learned mask | 60 | 250 | 40 |
| $in\_ch$ | Number of convolutional filters for input adaptation | 64 | 64 | 32 |
| $in\_ks$ | Kernel size of convolutional layer for input adaptation | 33 | 7 | 7 |
| $m\_ks$ | Kernel size of convolutional layers in the mask network | 7 | 3 | 7 |
| $m\_pool$ | Kernel size of max pooling layers in the mask network | [5,25] | [1,5] | [5,25] |
| $emb\_ks$ | Kernel size of convolutional layers in the embedding network | [33,33,7,7] | [7,7,7,7] | [7,7,7,7] |
| $emb\_st$ | Stride of convolutional layers in the embedding network | [5,5,2,2] | [1,2,5,5] | [2,2,2,1] |
| $n\_outputs$ | Number of final model outputs | 1 | 3 | 1 |
| | Dropout | $p = 0.2$ | disabled | disabled |
| | Loss | BCE with logits | MSE | BCE with logits |
| | Learning rate | 3e-4 | 1e-4 | 1e-4 |
| | Batch size | 32 | 64 | 64 |

*$L = \prod m\_pool$, $K \times L = T$.

