# OpenReview forum: "Learning Multiple Semantic Views For Self-explaining Physiological Signal Stratification"
_ICLR.cc/2025/Conference — Submitted to ICLR 2025_

### Official Review · Reviewer_9tMj · 2024-10-28

**Soundness:** 2
**Presentation:** 3
**Contribution:** 1
**Rating:** 5
**Confidence:** 4

**Summary:**

This paper presents an architecture for processing medical waveforms with enhanced explainability. The author claims the learned representations are task-relevant and human-interpretable.

**Strengths:**

The paper is generally easy to read, but its clarity can be enhanced by better diagrams.

**Weaknesses:**

1. In general, there is no significant technical innovation specially designed for clinical applications or medical waveforms. The author should explain how the proposed method differs from previous general xAI methods and compare the performances.
2. The author claims the method generates human-interpretable features, but the embedding and decision networks are not easily interpreted (limiting the transparency significantly).
3. There seems to be no user study with clinicians on the relevance of extracted features.
4. Figure 2 is too brief, consider adding some sub-figures to illustrate the ideas. It’s only about half the page width now.
5. Some equations on Page 5 seem un-necessary, and the notation can be simplified.
6. The tasks selected are not representative in general, and the SOTA methods cited are old in general.
7. The ablation is only limited to the number of views.
8. The results reported in Section 5.2 have strong selection bias (correctly-classified ones are shown).

**Questions:**

1. \[Line 16\] Can you explain how xAI “ensures” reliability without casual inference study?
2. \[Line 31\] Why do you believe validating on only 4 tasks with only 2 waveforms “displays universal usability”?
3. \[Line 37\] Is highlighting relevant regions sufficient for transparency? How does it relate to clinical decision making? Is there any assumption to be made here for how clinicians interact with the model you developed?
4. \[Line 328\] It seems 4-view is worse than 3-view. Is there any explanation?

---

> ### Author Response · Authors · 2024-11-23
>
> We thank reviewer 9tMj for providing detailed comments and suggestions, which have resulted in improving our work. Taking into account all reviewers’ comments and to better highlight the advantages of the proposed method, we have revised the manuscript by focusing on 2-view setting for 3 tasks, and conducting additional experiments to compare our method with the well-established post-hoc explanation methods (LIME and SHAP), both quantitatively and qualitatively.
>
> Below, we address each of the Reviewer’s comments.
>
> **4Q1: In general, there is no significant technical innovation specially designed for clinical applications or medical waveforms. The author should explain how the proposed method differs from previous general xAI methods and compare the performances.**
>
> > **4A1:** The use of ECG and PPG signals in our experiments is intended to demonstrate the applicability of the method to clinically relevant data.  To better demonstrate the explainability of the proposed model, in the revised manuscript, we added **Section 5.1** to quantitatively compare our method with two well-established post-hoc model interpretability methods, LIME and SHAP. The results suggest that the semantic segmentation masks, self-generated by the 2-view model, have better correspondence with the model’s testing performance, compared to explanations offered by LIME and SHAP.
>
> > Meanwhile, in the revised qualitative analysis (**Section 5.2**), our semantic masks have shown better consistency in highlighting important characteristics of the ECG and PPG signals related to the task, while the post-hoc model interpretability methods only found these characteristics in the signals occasionally.
> >
> > Moreover, the proposed self-explaining network is more efficient compared to post-hoc approaches, since it requires single model inference for generating explanations, while the post-hoc methods requires multiple model inferences to grasp the behavior of the model. We mentioned this in **Line 365**.
>
> **4Q2: The author claims the method generates human-interpretable features, but the embedding and decision networks are not easily interpreted (limiting the transparency significantly).**
>
> > **4A2:** We agree with the reviewer that the current architectures of the embedding and decision networks lack transparency. We think these could be addressed through using alternative model architectures, or using domain-agnostic model interpretation methods to describe the behavior of these networks, in combination with the interpretable representation generated by the semantic segmentation masks of our model. We leave these potential extensions of the current self-explaining model architecture for future works.
> >
> > This point is briefly addressed in **Line 517.**
>
> **4Q3: There seems to be no user study with clinicians on the relevance of extracted features.**
>
> > **4A3:** Please refer to our answer to **4Q11**.
>
> **4Q4: Figure 2 is too brief, consider adding some sub-figures to illustrate the ideas. It’s only about half the page width now.**
>
> > **4A4:** In the revised manuscript, we have accordingly re-formatted **Figure 2** with subfigures illustrating examples of masks and semantic views.
>
> **4Q5: Some equations on Page 5 seem un-necessary, and the notation can be simplified.**
>
> > **4A5:** We have accordingly removed the previous Equation 3, and added a short sentence describing the complementary relationship among the semantic states. We also replaced the notation $\mathbf{T}_n$ with the already-defined semantic states $\mathbf{u}_n$ in the revised **Equations (1),** **(2)** and **(6)** to simplify the notations.

---

> ### Author Response · Authors · 2024-11-23
>
> **4Q6: The tasks selected are not representative in general, and the SOTA methods cited are old in general.**
>
> > **4A6:** In this study, we considered two heterogeneous cardiovascular signals, and included both classification and regression tasks, to represent existing challenges in the field of cardiovascular signal stratification.  To better reflect the tasks considered in this work, we revised the title (in pdf) to emphasize cardiovascular signals.
> >
> > The SOTA methods cited in **Table 1** are state-of-the-art methods being strictly comparable with our method, as they were evaluated using the same training and testing sets employed in this work. We noted how we ensured such comparability in **Appendix A1**, and briefly mentioned in **Line 473**.
> >
> > We would like to note that for PPG-based HRV regression, although the QPPG and ERMA methods were proposed in 2018 and 2013, they are actively used in HRV studies. To better show the timeliness of these methods, we have updated their corresponding citations in **Table 1**, and mentioned them in **Line 475**.
>
> **4Q7: The ablation is only limited to the number of views.**
>
> > **4A7:** Our ablation study discussed the impact of removing the mask network from our multi-view model, which is the key architectural design of our multi-view network. We believe that this is the most essential mechanism in our model for offering self-interpretability, therefore, we formed our ablation study from this perspective.
>
> **4Q8: The results reported in Section 5.2 have strong selection bias (correctly-classified ones are shown).**
>
> > **4A8:** In the revised manuscript, in **Section 5.1**, we demonstrated the advantages of our self-explaining network in explainability, through quantitative analysis on testing sets including both correctly and wrongly-estimated samples. For the qualitative analysis, in **Section 5.2**, we stayed with correctly-classified samples, such that we can focus on comparing the alignment between interpretations created using different methods, with human’s clinical insights.
>
> **4Q9: [Line 16] Can you explain how xAI “ensures” reliability without casual inference study?**
>
> > **4A9:** The full sentence highlighted by the reviewer is “… such transparency is critical for ensuring reliability, identifying biases, …” The transparency offered by xAI technologies is indeed one of the critical factors that ensure reliability of deep learning models. We do not intent to imply that casual inference studies are excluded from xAI technologies.
> >
> > To highlight this point, we reworded this sentence in **Abstract** to “… such transparency is critical for **facilitating analysis of inference causality**, ensuring reliability, identifying biases, …”.
>
> **4Q10: [Line 31] Why do you believe validating on only 4 tasks with only 2 waveforms “displays universal usability”?**
>
> > **4A10:** In the revised manuscript, we included results from applying the proposed model on 3 different cardiovascular signal stratification tasks. These tasks demonstrated the broad applicability of the proposed model architecture, in terms of:
> >
> > 1. Being usable for analyzing two cardiovascular signals (ECG and PPG) which are characteristically different in waveform morphology and physiological information they provide.
> > 2. Being usable for both classification and regression tasks.
> > 3. Being usable for highlighting both beat-to-beat and regional patterns in the considered signals related to 3 different tasks.
> >
> > To describe these features more accurately, we have re-worded the term “**universal usability**” as “**broad applicability**”. Additionally, we revised the title (in pdf) to reflect that cardiovascular signals are considered in this work.
>
> **4Q11: [Line 37] Is highlighting relevant regions sufficient for transparency? How does it relate to clinical decision making? Is there any assumption to be made here for how clinicians interact with the model you developed?**
>
> > **4A11:** Although we took some insights from clinical studies of cardiovascular signals in our qualitative studies to interpret the semantic segmentation masks created by our model, at this proof-of-concept stage, the multi-view model is not designed to be used by clinicians. While clinical models should embed expert knowledge for alignment with decisions, our self-explaining model explores data-driven methods to uncover information from physiological signals.
> >
> > This point is briefly addressed in **Line 74.**
>
> **4Q12: [Line 328] It seems 4-view is worse than 3-view. Is there any explanation?**
>
> > **4A12:** The 4-view setting applies finer segmentation to the signal, which could have led the feature extraction network to learn from unimportant details in the signal. Since we have re-focused the paper on 2 views to provide more in-depth analysis of the proposed method, we leave further discussions on creating more semantic views for future work.

---

> > ### Comment · Reviewer_9tMj · 2024-12-02
> >
> > Thanks the authors for the detailed response. I increased my score based on the revisions.

---

### Official Review · Reviewer_mx35 · 2024-11-02

**Soundness:** 2
**Presentation:** 3
**Contribution:** 2
**Rating:** 5
**Confidence:** 4

**Summary:**

This paper introduces a self-explaining deep learning model architecture designed to enhance interpretability in the analysis of physiological signals, an issue often overlooked in existing deep models. The architecture employs a multi-semantic view approach, which generates multiple mask-modulated signal versions through a mask network. This process attributes model inputs to distinct semantic states, uncovering hidden patterns within the input data. The paper tests this architecture on four clinically relevant tasks involving ECG or PPG signals for classification and regression. Experimental results indicate that the multi-view approach demonstrates improved model interpretability, providing clearer insights into the model's decision-making process.

**Strengths:**

- The research topic of this paper focuses on the interpretability of medical artificial intelligence, which is a field of great concern and has significant practical importance.

- The method proposed in this paper maps different parts of the input signal to different semantic state spaces, revealing hidden patterns in the input signal that are related to model decisions, thereby enhancing the model's interpretability.

- This paper has been validated on multiple datasets, and the experimental results show that the model's decision focus aligns with domain knowledge, verifying the effectiveness of the method.

**Weaknesses:**

- Data diversity: The dataset used in this article is limited to a single type of physiological signal, utilizing either ECG or PPG signals exclusively (although differentiated PPG signals were employed). This limitation raises questions about the effectiveness of the proposed method when applied to mixed types of physiological signal inputs, which warrants further validation.
- Semantic state complexity: The number of semantic states in the paper is relatively small (2, 3, or 4). For complex inputs or tasks, a limited number of semantic states may not adequately reflect the model's decision-making process. The performance of the model with a higher number of semantic states requires further investigation.
- Visualization challenge: The visualization results are discernible when the number of semantic states is small (e.g., 2). However, as the number of semantic states increases, these visualizations become difficult to recognize effectively. This can lead to a decreased understanding of the model's decision focus, thereby reducing the model's interpretability.
- Evaluation metrics: There are concerns with the evaluation metrics used in the dataset. In classification problems, accuracy is employed as the evaluation metric, but this metric is susceptible to the impact of class imbalance. More robust metrics, such as AUC or F-score, should be considered for a more reliable assessment.
- Temporal data representation: The method proposed in the paper classifies semantic states for individual sample time points. However, data from a single time point may not capture sufficient semantic information, especially when the signal sampling frequency is high, which could limit the representation of meaningful physiological information.

**Questions:**

See above

---

> ### Author Response · Authors · 2024-11-23
>
> We thank reviewer mx35 for providing detailed comments and suggestions, which have resulted in improving our work. Taking into account all reviewers’ comments and to better highlight the advantages of the proposed method, we have revised the manuscript by focusing on 2-view setting for 3 tasks, and conducting additional experiments to compare our method with the well-established post-hoc explanation methods (LIME and SHAP), both quantitatively and qualitatively.
>
> Below, we address each of the Reviewer’s comments.
>
> **3Q1: [Data diversity] The dataset used in this article is limited to a single type of physiological signal, utilizing either ECG or PPG signals exclusively (although differentiated PPG signals were employed). This limitation raises questions about the effectiveness of the proposed method when applied to mixed types of physiological signal inputs, which warrants further validation.**
>
> > **3A1:** Interpreting deep learning models taking mixed types of physiological signals as input can be intrinsically challenging. For example, ECG and PPG signals are characteristically different in waveform morphology and physiological information, and there are intrinsic delays between the two signals due to the time required for pulse transition. These heterogeneities may raise questions to whether it is feasible or meaningful to create interpretations for mixed physiological signal inputs. As such, the applicability of our method to mixed physiological signals is outside the scope of our study.
> >
> > As this work considered tasks involving PPG and ECG, we revised the title (in pdf) to reflect that cardiovascular signals are considered in this study.
>
> **3Q2: [Semantic state complexity] The number of semantic states in the paper is relatively small (2, 3, or 4). For complex inputs or tasks, a limited number of semantic states may not adequately reflect the model's decision-making process. The performance of the model with a higher number of semantic states requires further investigation.**
>
> > **3A2:** The objective of our multi-view model is to create human-understandable explanations on its own, for highlighting informative patterns in the physiological signal that drives model’s output. The number of semantic views is a hyperparameter, selected to optimize the correctness, sensitivity, and human understandability of the learned interpretations. In the revised manuscript, to include quantitative interpretability analysis, we have re-focused the paper to 2-views, and left discussions on learning more semantic views for future studies. We mentioned this in **Line 278** of the revised manuscript.
>
> **3Q3 [Visualization challenge] The visualization results are discernible when the number of semantic states is small (e.g., 2). However, as the number of semantic states increases, these visualizations become difficult to recognize effectively. This can lead to a decreased understanding of the model's decision focus, thereby reducing the model's interpretability.**
>
> > **3A3:** In the revised manuscript, to include quantitative interpretability analysis, we re-focused the paper on 2-views, and left discussions on multiple view for future work. With respect to the difficulties in inspecting visualizations, we have accordingly updated **Figures 4-6**, for better clarity in visualizing the learned semantic masks.
> >
> > Moreover, we formed quantitative analysis, which has shown that our results outperform interpretations generated through well-established post-hoc counterparts, such as LIME and SHAP.
>
> **3Q4: [Evaluation metrics] There are concerns with the evaluation metrics used in the dataset. In classification problems, accuracy is employed as the evaluation metric, but this metric is susceptible to the impact of class imbalance. More robust metrics, such as AUC or F-score, should be considered for a more reliable assessment.**
>
> > **3A4:** In the revised manuscript, we have accordingly included AUC and F1 metrics for the OSA and $\Delta$BP classification tasks, in **Table 1**.
>
> **3Q5: [Temporal data representation] The method proposed in the paper classifies semantic states for individual sample time points. However, data from a single time point may not capture sufficient semantic information, especially when the signal sampling frequency is high, which could limit the representation of meaningful physiological information.**
>
> > **3A5:** To prevent attributing discreate samples into semantic states, we enforced a minimum window duration $L$ in our mask network, within which all samples are attributed to the same semantic state. We mentioned this at **Line 231**, discussed its implementation in detail in **Equation (6)-(7)** and **Line 931**, and listed selections of this parameter for each considered task in **Table 2** (in **Appendix**).

---

> > ### Comment · Reviewer_mx35 · 2024-11-28
> >
> > I appreciate the authors' feedback. After reviewing their comments, I've concluded that my initial score stands as is.

---

> > > ### Author Response · Authors · 2024-12-01
> > >
> > > Thank you for taking the time to review our responses and the improvements we have made. We would like to highlight an additional change we made to the paper. While the primary objective of this paper was to improve explainability, we also explored the possibility of further improving task-level performance by incorporating more advanced blocks into the model, as suggested by **Reviewer zyVj**. We conducted an additional experiment for the $\Delta$BP classification task by replacing the ResNet block in our 2-view model with Res2Net [1], a more advanced variation of ResNet. Our results demonstrated that this substitution, indeed offered an improvement on the task-level performance from **0.729** to **0.739** in F1 score, suggesting the potential for further improving the task-level performance of our 2-view model by incorporating more advanced blocks. We have now indicated this point briefly in **Line 505** of our latest revision.
> > >
> > > [1] Shang-Hua Gao, et al. Res2Net: A New Multi-Scale Backbone Architecture. *IEEE TPAMI*, 2019.

---

### Official Review · Reviewer_5z4n · 2024-11-02

**Soundness:** 2
**Presentation:** 3
**Contribution:** 2
**Rating:** 6
**Confidence:** 4

**Summary:**

The work involves an inherently explainable AI approach for clinically relevant supervised tasks based on electrocardiogram (ECG) or photoplethysmogram (PPG) inputs. The explainability is achieved by exploiting trainable masks to identify regions of ECG/PPG contributing to clinically significant information.

**Strengths:**

This work presents an original idea for a generalized explainable deep learning architecture that can potentially have significant implications for AI-based medicine and XAI. Specifically, it incorporates model and sample level interpretability, by introducing a prior constraint on the number of semantic views which are trainable (model level explainability), based on which each sample produces a unique segmentation mask (sample level explainability). This is contrary to post hoc explainability techniques, where the sample-level explanation can be independent, ambiguous (model approximations), and inconsistent across techniques.

**Weaknesses:**

The proposed deep learning architecture can be relevant for signal segmentation tasks but the level of explainability is quite coarse. The method involves qualitative (visual) exploration of the learned masks that may be quite difficult for slightly more complex tasks. This is also evident from the fact that performance is only optimum for 2 or 3 masks, with the design preventing the integration of multi-dimensional concepts. Moreover, the learned views do not necessarily seem to provide unique insights into model explainability, without knowing which semantic states, and which part of the signals in each mask, contribute to the networks’ decisions (e.g., in the example of AF, the presence or absence of P waves may be more indicative for the detection of the disease, than actual QRS peaks). The selection of tasks is also quite limited in showing interpretability properties, considering that all tasks in the paper are defined as conditions related to peak-to-peak interval variability.

**Questions:**

Abstract

Methods in the Abstract do not need to be so extensive e.g., The section: “Specifically, the proposed network… to the task labels” could be omitted.

Methods

Line 181. “… each sample x in the time series S to be attributed to one of the N semantic states”.
Why should one sample be attributed to only one semantic state? Intuitively, it seems that specific parts of a physiological signal could reflect several latent semantic states.

Equation (4). By applying softmax activation at each time sample, we allow the information to leak into all masks with varying amplitudes, which deviates from the original strict binary definition. In this case, how do we know whether this information is amplified or suppressed by the subsequent embedding networks – and hence attribute explainability to the high-amplitude parts?

3.2.2. Considering that the semantic segmentation masks provide interpretability, why do we need weight sharing in the embedding network features?

3.2.2. What’s the role of the differential embedding vector? Is there any empirical evidence that the decision network can’t exploit such relationships?

Results

Table 1. It would be preferable to report AUC metrics instead of accuracy, considering that accuracy will be sensitive to each model and binary threshold (was there any threshold tuning?)

Line 337. “which suggests the effectiveness… from full-time series interval”. The comparison here may not be fair, considering that removing the mask network significantly reduces the number of parameters in the model, which will solely rely on one embedding network to receive input from the original signal.

Line 370. “From Figure 3… clearly capture such information”. The heart rate variation is not very prominent in the figures. Maybe you could show smaller windows or wider X-axes?

Appendix

Line 926. “we enforce a minimum duration L”. How did you select L for each task? I don’t think these numbers are mentioned in the paper.

General

Please generate in-text citations with brackets.

From a physiological signal interpretation perspective, how do these semantic views compare to existing post-hoc explainability methods? E.g., clustering techniques at the sample level [1].

Potential Work

The assumptions behind the semantic masks (semantic states attributed to specific time samples) and the need for prior selection of the number of semantic states N may introduce limitations as a general-purpose explainability mechanism. Could the network somehow discover the optimal number of semantic states? (instead of predefining N).

References:

[1]  Boubekki, A., Fadel, S.G., & Mair, S. (2024). Leveraging Activations for Superpixel Explanations. ArXiv, abs/2406.04933.

---

> ### Author Response · Authors · 2024-11-23
>
> We thank reviewer 5z4n for leaving detailed comments and suggestions, which have resulted in improving our work. Taking into account all reviewers’ comments and to better highlight the advantages of the proposed method, we have revised the manuscript by focusing on 2-view setting for 3 tasks, and conducting additional experiments to compare our method with the well-established post-hoc explanation methods (LIME and SHAP), both quantitatively and qualitatively.
>
> Below, we address each of the Reviewer’s comments.
>
> **2Q1: The proposed deep learning architecture can be relevant for signal segmentation tasks but the level of explainability is quite coarse. The method involves qualitative (visual) exploration of the learned masks that may be quite difficult for slightly more complex tasks. This is also evident from the fact that performance is only optimum for 2 or 3 masks, with the design preventing the integration of multi-dimensional concepts.**
>
> > **2A1:** To address this comment, in the revised manuscript, we added **Section 5.1** to quantitatively compare the interpretability of our method with post-hoc explanation methods, LIME and SHAP. The results suggest that the semantic segmentation masks, self-generated by the 2-view model, have better correspondence with the model’s testing performance, compared to explanations offered by LIME and SHAP. Please note that the revised manuscript now only focuses on 2 semantic views to enable a more in-depth analysis and performance evaluation of the proposed approach. We have decided to leave discussions on learning additional semantic views for future work.
>
> **2Q2: Moreover, the learned views do not necessarily seem to provide unique insights into model explainability, without knowing which semantic states, and which part of the signals in each mask, contribute to the networks’ decisions (e.g., in the example of AF, the presence or absence of P waves may be more indicative for the detection of the disease, than actual QRS peaks). The selection of tasks is also quite limited in showing interpretability properties, considering that all tasks in the paper are defined as conditions related to peak-to-peak interval variability.**
>
> > **2A2:** To provide more clarification and better demonstrate the explainability of the proposed model, in the revised manuscript, we both quantitatively (**Section 5.1**) and qualitatively (**Section 5.2**) compare the performance of our proposed model with two well-established post-hoc explanation methods, LIME and SHAP.
> >
> > Quantitatively, results show that regions with high amplitudes in one or more semantic masks have the greatest impact on the model’s output. The sensitivity of these regions in affecting the model’s performances, outperforms regions suggested by LIME or SHAP.
> >
> > Qualitatively, it can be observed that our semantic masks have better consistency in highlighting important characteristics of the ECG and PPG signals, while LIME or SHAP only found these characteristics in the signals occasionally.
> >
> > We respectfully argue that not all tasks in the paper are only defined as conditions related to peak-to-peak interval variability. For example, the $\Delta$BP classification task is defined as estimating changes in amplitude of the arterial blood pressure, based on the morphology of another physiological signal (the PPG). The change in arterial blood pressure amplitude has no direct connection with the peak-to-peak interval of the PPG signal. For example, in subfigure (c2) of **Figure 6**, there are no major differences between the inter-peak intervals of the PPG signal. The semantic masks created by our model successfully locate morphological changes in higher-order derivatives of the PPG signal, which align with the changes in reference BP values (marked on the top of column (2) of **Figure 6**).
> >
> > Additionally, as this work considered tasks involving PPG and ECG, we updated the title (in pdf) to emphasize cardiovascular signals. Due to space limitations, we also removed the discussions on the AF task.
>
> **2Q3: [Abstract] Methods in the Abstract do not need to be so extensive e.g., The section: “Specifically, the proposed network… to the task labels” could be omitted.**
>
> > **2A3:** We accordingly updated the abstract.
>
> **2Q4: [Line 181] “… each sample x in the time series S to be attributed to one of the N semantic states”. Why should one sample be attributed to only one semantic state? Intuitively, it seems that specific parts of a physiological signal could reflect several latent semantic states.**
>
> > **2A4:** While it is possible to develop our multi-view model to attribute a sample in the time series to multiple semantic states, we enforced the single semantic state assumption to regularize our model toward using distinct borderlines to segment the time series into multiple regions. This enhances the practicality and easiness for humans to interpret the learned semantic views. We emphasized this point in **Line 157**.

---

> ### Author Response · Authors · 2024-11-23
>
> **2Q5: [Equation (4)] By applying softmax activation at each time sample, we allow the information to leak into all masks with varying amplitudes, which deviates from the original strict binary definition. In this case, how do we know whether this information is amplified or suppressed by the subsequent embedding networks – and hence attribute explainability to the high-amplitude parts?**
>
> > **2A5:** Thank you for this comment. Indeed, this is a limitation of the current multi-view model, since the embedding and decision networks themselves lack integrated interpretability. For future improvements and addressing this issue, alternative architectures for the embedding network for better interpretability can be considered, or domain-agnostic model interpretation methods to describe the behavior of the embedding network can be leveraged. We leave these potential extensions of the current self-explaining model architecture for future works. This point is addressed in **Line 517**.
>
> **2Q6: [3.2.2] Considering that the semantic segmentation masks provide interpretability, why do we need weight sharing in the embedding network features?**
>
> > **2A6:** The use of shared embedding network is essential to prevent our model from turning into a trivial network that generates no segmentation. Empirically, we found that during the early stage of training, the mask network might assign all samples to only one semantic view. If separated embedding networks are used for feature extraction from each semantic views, the embedding networks of other semantic states will receive (nearly) zeroed input. As a result, these embedding networks find it difficult to propagate informative gradient to the mask network, preventing updates the zeroed semantic states and valid segmentations of the input time series. When using weight sharing, this problem is addressed, since the embedding network can always be updated by the gradient of any non-zero semantic state, which drives the mask network to create proper segmentations.
> >
> > This point is briefly addressed in **Line 259**.
>
> **2Q7: [3.2.2] What’s the role of the differential embedding vector? Is there any empirical evidence that the decision network can’t exploit such relationships?**
>
> > **2A7:** We have removed all related descriptions to this setting in the revised manuscript, since it is not a general setting that adds to the self-explainability  of our model discussed in this study.
>
> **2Q8: [Table 1] It would be preferable to report AUC metrics instead of accuracy, considering that accuracy will be sensitive to each model and binary threshold (was there any threshold tuning?)**
>
> >**2A8:** Thank you for this comment. In the revised manuscript, we have additionally included AUC and F1 metrics for the OSA and $\Delta$BP classification tasks, to address the limitations of reporting ACC only.
> >
> >These two considered binary classification tasks do not involve threshold tuning. We simply train the model to produce logits and use 0 as the threshold.
>
> **2Q9: [Line 337] “which suggests the effectiveness… from full-time series interval”. The comparison here may not be fair, considering that removing the mask network significantly reduces the number of parameters in the model, which will solely rely on one embedding network to receive input from the original signal.**
>
> > **2A9:** Please note that in contrast to traditional feed-forward neural networks, in which earlier modules output high-dimensional embeddings as inputs for later modules, our multi-view model architecture employs an embedding network that only receives raw physiological signals, segmented by the mask network in different ways, for generating model’s output. As such, the difference in the input to the embedding networks in the full and ablation models, is limited to the presence or absence of mask modulation. Moreover, due to weight sharing in the embedding network, the number of parameters in the embedding networks of the full and ablation models is exactly the same, and the only difference is in the number of parameters of the decision network, due to combining multiple embedding vectors in the full model. As such, the comparison between the full and ablation models indeed highlights the effectiveness of learning from the same input time series using multiple complementary perspectives.
> >
> > This point is briefly addressed in **Line 468** and **Line 480**.

---

> ### Author Response · Authors · 2024-11-23
>
> **2Q10: [Line 370] “From Figure 3… clearly capture such information”. The heart rate variation is not very prominent in the figures. Maybe you could show smaller windows or wider X-axes?**
>
> > **2A10:** We have revised this figure (now **Figure 4** in the revised manuscript) to improve the readability and better highlight the heart rate variation.
>
> **2Q11: [Line 926] “we enforce a minimum duration L”. How did you select L for each task? I don’t think these numbers are mentioned in the paper.**
>
> > **2A11:** We apologize for lack of clarity in the previous version. $L$ is selected manually through trial and error experiments for optimizing the human understandability and consecutiveness of the semantic segmentation masks. $L$ is controlled by the kernel sizes of max pooling layers in the mask network (**Figure 7** in **Appendix**). For better explanation, in the revised manuscript, we mentioned $L$ in **Line 232**, and noted its value and relationship with other model parameters for all tasks (**Line 931** and **Table 2**, in **Appendix**).
>
> **2Q12: [General] Please generate in-text citations with brackets.**
>
> > **2A12:** We have updated all in-text citations with brackets in the revised manuscript.
>
> **2Q13: [General] From a physiological signal interpretation perspective, how do these semantic views compare to existing post-hoc explainability methods? E.g., clustering techniques at the sample level [1].**
>
> > **2A13:** Thank you for this comment. In the revised manuscript, we present quantitative (**Section 5.1**) and qualitative (**Section 5.2**) comparison with well-established post-hoc model interpretation methods, LIME and SHAP. Quantitative comparison results suggest that the semantic segmentation masks, self-generated by the 2-view model, have better correspondence with the model’s testing performance, compared to the post-hoc methods. From qualitative analysis (**Section 5.2**), we found LIME and SHAP to fall short in capturing key task-related characteristics of the ECG and PPG signal over all beats, compared to the semantic masks self-generated by our model.
>
> **2Q14 [Potential Work] The assumptions behind the semantic masks (semantic states attributed to specific time samples) and the need for prior selection of the number of semantic states N may introduce limitations as a general-purpose explainability mechanism. Could the network somehow discover the optimal number of semantic states? (instead of predefining N).**
>
> > **2A14:** In general, we believe that $N$ can be optimized through hyperparameter sweep. An optimized $N$ should provide best model performance on the desired tasks, while yielding interpretations that are human-understandable, and have strong correlation with model’s behavior. Since we have re-focused the paper on 2-views, we leave further investigations to creating more than 2 views for future work.

---

> > ### Comment · Reviewer_5z4n · 2024-11-27
> > **Further comments**
> >
> > Thanks for the additional work but the result for only 2 masks is not very informative. It seems that the performance will likely degrade with more complex concepts.

---

> > > ### Author Response · Authors · 2024-12-01
> > >
> > > Thank you for taking the time to review our responses and the improvements we have made. While we understand your concern, we believe that using 2 masks can still provide informative and critical insights depending on the task, as demonstrated in our results, in particular for the OSA classification task, and the $\Delta$BP classification task. Furthermore, the results from our 2-view model also show quantitative and qualitative performance improvement compared to well-established explainability techniques.
> > >
> > > This work serves as the baseline for introducing our proposed self-explaining multi-view deep learning architecture. In our earlier draft, we showed that additional views are possible, but in this latest version, we focused the experimental results on 2-view in order to comprehensively compare the performance of the proposed approach with other xAI techniques, both quantitatively and qualitatively. Due to space constraints, we are just not able to include results and comprehensive discussions for other views. We believe this can be addressed in future work.
> > >
> > > We would like to also highlight an additional change we made to the paper. While the primary objective of this paper was to improve explainability, we also explored the possibility of further improving task-level performance by incorporating more advanced blocks into the model, as suggested by **Reviewer zyVj**. We conducted an additional experiment for the $\Delta$BP classification task by replacing the ResNet block in our 2-view model with Res2Net [1], a more advanced variation of ResNet. Our results demonstrated that this substitution, indeed offered an improvement on the task-level performance from **0.729** to **0.739** in F1 score, suggesting the potential for further improving the task-level performance of our 2-view model by incorporating more advanced blocks. We have now indicated this point briefly in **Line 505** of our latest revision.
> > >
> > > [1] Shang-Hua Gao, et al. Res2Net: A New Multi-Scale Backbone Architecture. *IEEE TPAMI*, 2019.

---

### Official Review · Reviewer_zyVj · 2024-11-04

**Soundness:** 2
**Presentation:** 3
**Contribution:** 2
**Rating:** 6
**Confidence:** 3

**Summary:**

This paper presents a multi-view deep learning model aimed at self-explaining predictions for various physiological signal-based tasks, such as obstructive sleep apnea (OSA) and atrial fibrillation (AF) detection. The proposed model generates “semantic views” by using mask networks to isolate task-relevant regions of the input signals. These views are used to enhance interpretability and yield clinically relevant insights.

**Strengths:**

1- The multi-view segmentation approach is an innovative contribution, adding potential value to explainability in machine learning for healthcare, where interpretability is crucial.

2- The paper proposes a unified architecture applicable to both classification and regression tasks, which shows adaptability to a variety of physiological signal processing tasks.

**Weaknesses:**

1- The paper introduces multiple semantic views (2, 3, or 4 views) but does not explain why these specific numbers of views are optimal across tasks. This arbitrary choice may limit the interpretability and generalizability of the approach. Further discussion or empirical testing regarding the impact of varying the number of views on interpretability and performance would strengthen the approach.

2- The experimental setup could have greatly benefited from ablation studies that justify the architectural decisions, such as the number of mask networks or the use of shared embedding networks. These studies would help clarify the impact of each component on the model’s performance and interpretability, providing a stronger empirical basis for the architectural choices.

3- The authors claim alignment between the generated views and clinical knowledge, yet this is primarily presented through visual inspection. Providing more robust, quantitative evaluations of interpretability, ideally verified with domain experts, would lend credibility to these claims.

4-  The results are not entirely convincing, as the proposed model fails to outperform current state-of-the-art implementations on 3 out of 4 datasets. Additionally, the ablation study in Table 1 indicates that the multi-view architecture offers only a marginal performance improvement.

5- There is a lack of comparison with established explainability approaches like SHAP or LIME. Although these methods may not offer the same level of task-specific interpretability, a comparison would clarify the relative benefits of the proposed model.

Minor:

1- The naming conventions in Table 1 could be clearer. Terms like “SOTA” and “ablation” could be replaced with more descriptive labels that specify the method or configuration used, making it easier for readers to understand the comparison.

**Questions:**

Major:

See above ("Weaknesses").

Minor:

1- Given the reliance on task labels to optimize segmentation, how does the model perform on tasks with sparse or noisy labels? Does this affect interpretability? It would be interesting if authors could've addressed that and potentially compare it with the SOTA method.

---

> ### Author Response · Authors · 2024-11-23
>
> We thank Reviewer zyVj for the constructive comments and suggestions, which have resulted in improving our work. Taking into account all reviewers’ comments and to better highlight the advantages of the proposed method, we have revised the manuscript by focusing on 2-view setting for 3 tasks, and conducting additional experiments to compare our method with the well-established post-hoc explanation methods (LIME and SHAP), both quantitatively and qualitatively.
>
> Below, we address each of the Reviewer’s comments.
>
> **1Q1: The paper introduces multiple semantic views (2, 3, or 4 views) but does not explain why these specific numbers of views are optimal across tasks. This arbitrary choice may limit the interpretability and generalizability of the approach. Further discussion or empirical testing regarding the impact of varying the number of views on interpretability and performance would strengthen the approach.**
>
> > **1A1:** To better highlight the advantages of the proposed method compared to other well-established methods, both qualitatively and quantitatively, the revised manuscript only considers 2 semantic views. We have decided to leave discussions on learning additional semantic views for future work. We have mentioned this in **Line 279** of the revised manuscript. We would like to add that the number of semantic views is a hyperparameter, selected to optimize the correctness, sensitivity, and human understandability of the learned interpretations.
>
> **1Q2: The experimental setup could have greatly benefited from ablation studies that justify the architectural decisions, such as the number of mask networks or the use of shared embedding networks. These studies would help clarify the impact of each component on the model’s performance and interpretability, providing a stronger empirical basis for the architectural choices.**
>
> > **1A2:** Thank you for this comment. As for the number of masks, we have re-focused the revised manuscript on 2-views and left further investigations into creating more than 2 semantic views for future work.
> >
> > The use of shared embedding network is essential to prevent our model from turning into a trivial network that generates no segmentation. Empirically, we found that during the early stage of training, the mask network might assign all samples to only one semantic view. If separated embedding networks are used for feature extraction from each semantic view, the embedding networks of other semantic states will receive (nearly) zeroed input. As a result, these embedding networks find it difficult to propagate informative gradient to the mask network, preventing updates to zeroed semantic states and valid segmentations of the input signal. When using weight sharing, this problem is addressed, since the embedding network can always be updated by the gradient of any non-zero semantic state, which drives the mask network to create proper segmentations.  This point is briefly addressed in **Line 259**.
>
> **1Q3: The authors claim alignment between the generated views and clinical knowledge, yet this is primarily presented through visual inspection. Providing more robust, quantitative evaluations of interpretability, ideally verified with domain experts, would lend credibility to these claims.**
>
> > **1A3:** Please refer to our answer to **1Q5**. We have included quantitative analysis in the revised manuscript.
>
> **1Q4: The results are not entirely convincing, as the proposed model fails to outperform current state-of-the-art implementations on 3 out of 4 datasets. Additionally, the ablation study in Table 1 indicates that the multi-view architecture offers only a marginal performance improvement.**
>
> > **1A4:** Our main focus in this work was to showcase the interpretability of the learned masks. As a proof-of-concept study, we only utilized very basic deep learning architectures (CNN, LSTM and MLP), to emphasize the architectural design of the multi-view model for enabling self-explainability. Many modules in this model can be replaced with more advanced alternatives, which can potentially improve the classification / regression performance further. For example, the mask network may be replaced by a fully convolutional network (e.g., U-Net) or other advanced time series segmentation architectures. We leave further explorations on the implementation of our multi-view model architecture for future works. This point is briefly addressed in **Line 503**.

---

> > ### Comment · Reviewer_zyVj · 2024-11-24
> >
> > I would like to thank the authors for their detailed responses and for addressing all the questions thoroughly. While I appreciate the objective of the work, which is to emphasize explainability, I believe that, ultimately, the community is also interested in understanding whether the proposed approach can advance the state-of-the-art.
> >
> > Furthermore, the claim that substituting components of the model with more complex alternatives would yield better results is quite bold and lacks both experimental evidence and theoretical justification. As we know, such improvements are not always linear; in fact, simpler methods often lead to enhanced explainability and robustness.
> >
> > Based on the provided information, I will increase my score to 6. However, I must lower my confidence level as I remain uncertain whether the contributions and results are entirely suitable for the ICLR audience.

---

> > > ### Author Response · Authors · 2024-12-01
> > >
> > > Thank you very much for your positive comments on our revised version. With respect to your concern on whether using more complex components in the model would yield better results, we conducted an additional experiment for the $\Delta$BP classification task by replacing the ResNet block in our 2-view model with Res2Net [1], a more advanced variation of ResNet. Our results demonstrated that this substitution, indeed offered an improvement on the task-level performance from **0.729** to **0.739** in F1 score, suggesting the potential for further improving the task-level performance of our 2-view model by incorporating more advanced blocks. We have now indicated this point briefly in **Line 505** of our latest revision.
> > >
> > > [1] Shang-Hua Gao, et al. Res2Net: A New Multi-Scale Backbone Architecture. *IEEE TPAMI*, 2019.

---

> ### Author Response · Authors · 2024-11-23
>
> **1Q5: There is a lack of comparison with established explainability approaches like SHAP or LIME. Although these methods may not offer the same level of task-specific interpretability, a comparison would clarify the relative benefits of the proposed model.**
>
> > **1A5:** Thank you for this suggestion. In the revised manuscript, we added **Section 5.1** to quantitatively compare the interpretability of our method with post-hoc explanation methods, LIME and SHAP. The results suggest that the semantic segmentation masks, self-generated by the 2-view model, have better correspondence with the model’s testing performance, compared to explanations offered by LIME and SHAP.
>
> **1Q6: The naming conventions in Table 1 could be clearer. Terms like “SOTA” and “ablation” could be replaced with more descriptive labels that specify the method or configuration used, making it easier for readers to understand the comparison.**
>
> > **1A6:** We have accordingly revised **Table 1** and noted each method with a short descriptive label.
>
> **1Q7: Given the reliance on task labels to optimize segmentation, how does the model perform on tasks with sparse or noisy labels? Does this affect interpretability? It would be interesting if authors could've addressed that and potentially compare it with the SOTA method.**
>
> > **1A7:** Investigating the interpretable representations when training our network on sparse or noisy labels would be an interesting topic to explore. Since our model can learn different segmentations for different task labels, we expect the perturbed labels to also affect the semantic masks learned by our model. Given the focus of this work and space constraints, we leave further exploration on this topic for future studies.

---

### Author Response · Authors · 2024-12-04
**Post-Rebuttal Summary**

The authors thank all the reviewers for their time, constructive comments and suggestions during the rebuttal period. Below we would like to provide a summary of the rebuttal discussions and an overview of the revisions that were made accordingly to address all the concerns and questions raised during the rebuttal period.

**Goal of the work:** This XAI-focused paper presents a new self-explaining multi-view deep learning architecture that generates task-relevant, human-interpretable masks to highlight feature importance during model inference. As a proof-of-concept, the 2-view version of this architecture is evaluated for three different clinically-relevant tasks involving cardiovascular signals. Quantitative and qualitative results show that the proposed model's self-generated masks outperform established post-hoc methods (LIME and SHAP) in explainability for these various tasks. Furthermore, the model, implemented with basic networks, achieves task-level performance that is comparable or better than the state-of-the-art methods. Using advanced alternatives improves task-level performance further.

**Strengths:** The reviewers

- **[Contribution]** acknowledged the innovative idea and original contribution of the proposed generalized multi-view architecture for enhancing explainability in healthcare AI,
- **[Impact]** recognized the importance of the work, and its potential impact in healthcare AI, where interpretability can be crucial,
- **[Novelty]** noted that by producing a unique segmentation mask for each sample, the proposed method addresses limitations of post-hoc explainability techniques that provide independent, ambiguous, and inconsistent explanations,
- **[Adaptability]** commented on the adaptability of the proposed architecture in handling both classification and regression tasks, and its applicability to a variety of physiological signal processing tasks,
- **[Experiments]** noted that experiments using various datasets demonstrate that the model's decision focus aligns with domain knowledge, confirming its effectiveness, and
- **[Readability]** rated that the paper is clearly presented and easy to read.

**Concerns:** The reviewers’ major comments and questions can be generally categorized as follows:

- providing quantitative and qualitative comparison with other XAI methods,
- clarifying the necessity of the architectural design, number of semantic states/views and temporal data representation,
- task-level performance, and including other evaluation metrics, and
- improving visualization and the clarity of figures/tables.

**Revisions:** Taking all reviewers’ comments into account, we have made major revisions to our manuscript (highlighted in blue in the updated PDF) and addressed all the comments. In summary:

- **[Quantitative and Qualitative Analysis]** We conducted quantitative and qualitative performance comparisons of the proposed approach with two well-established post-hoc explanation methods (LIME and SHAP). The model's self-generated explanations outperform LIME and SHAP in correctness, sensitivity, and efficiency, while also qualitatively showing better consistency in capturing clinically relevant cardiovascular patterns.
- **[Necessity of the Architectural Design]** We provided clarifications and explanations on the key architectural designs, including weight sharing in the model's embedding network and the number of views generated. For a more in-depth analysis, we focused on the 2-view model in the experimental results, to highlight the most discernible patterns in cardiovascular signals and to provide comprehensive comparisons with other XAl techniques. Although the feasibility of generating more than 2 views was shown in our initial submission, we choose to address them in future works.
- **[Temporal Data Representation]** We further explained the implementation of minimum duration $L$ in our semantic masks that enforces informative segments, instead of single time points, to be preserved in each semantic state.
- **[Task-level Performance]** We included additional performance metrics (AUC and F1 score) to enable a more comprehensive task-level performance comparison. Additionally, while the primary objective of this paper was to improve explainability, we also demonstrated the possibility of further improving task-level performance by incorporating more advanced blocks into the model.
- **[Visualizations/Figures/Tables Clarity]** We updated the tables and figures to better clarify the main concepts, semantic segmentation masks, explainability comparisons, task-level performance, and parameter settings for each task.

**Post-revision comments:** All reviewers recognized our efforts in addressing their comments and questions, with some emphasizing the detailed and thorough nature of our responses. 3 out of 4 reviewers raised their scores.

---

### Meta-Review · Area_Chair_AyTR · 2024-12-20

**Metareview:**

This paper introduces a multi-view deep learning model aimed at self-explaining predictions for various physiological signal-based tasks. The proposed model generates “semantic views” by using mask networks to isolate task-relevant regions of the input signals. These views are used to enhance interpretability and yield clinically relevant insights. Experiments are conducted on regression and classification tasks related to cardiovascular health, involving electrocardiogram (ECG) or photoplethysmogram (PPG) signals. The paper is about AI for healthcare, well-written, and the interpretability of medical problems is an important topic. From the methodology perspective, the embedding and decision networks are not easily interpreted, so the overall transparency is still limited, which undermines the authors' claim on interpretability. The current architectures of the embedding and decision networks need more transparency. The paper needs some discussions on how the method helps the decision-making by clinicians. From the application perspective, there is no significant innovation specially designed for clinical applications or medical waveforms.

**Additional Comments On Reviewer Discussion:**

There are detailed discussions between the reviewers and authors. Since this paper is really a borderline one, the AC calls the discussions among reviewers and AC. There is no reviewer championing the paper. All reviewers agreed that there are clear limitations in this work.

---

### Decision · Program_Chairs · 2025-01-22

Reject